# Future Perspective of Chemotherapy and Pharmacotherapy in Thymic Carcinoma

**DOI:** 10.3390/cancers13205239

**Published:** 2021-10-19

**Authors:** Rui Kitadai, Yusuke Okuma

**Affiliations:** 1Department of Thoracic Oncology and Respiratory Medicine, Tokyo Metropolitan Cancer and Infectious Diseases Center Komagome Hospital, Tokyo 113-8677, Japan; rkitadai@cick.jp; 2Department of Thoracic Oncology, National Cancer Center Hospital, Tokyo 104-0045, Japan

**Keywords:** thymic carcinoma, chemotherapy, molecular targeted agent, immune checkpoint inhibitor, clinical trial

## Abstract

**Simple Summary:**

Thymic carcinoma is a rare cancer, and its biology remains largely unknown. Although complete surgical resection is a standard treatment for thymic carcinoma, systemic chemotherapy is frequently administered in metastatic or recurrent cases. Given the rarity, therapeutic agents are often confirmed on the basis of the results of phase II trials or retrospective studies. Platinum-based combination chemotherapy has long been employed for treating thymic carcinoma. Recently, biomarkers have been explored, and molecular profiles and major oncogenic pathways have gradually been revealed by next-generation sequencing, resulting in the development of targeted therapies. Moreover, clinical trials assessing combination therapy with immune checkpoint inhibitors are ongoing and are expected to be efficacious for treating thymic epithelial tumors. We reviewed the current role of systemic chemotherapy, including targeted therapies and immune checkpoint inhibitors, considering recent findings regarding its biology.

**Abstract:**

Thymic carcinoma is a rare cancer that arises from thymic epithelial cells. Its nature and pathology differ from that of benign thymoma, presenting a poorer prognosis. If surgically resectable, surgery alone or surgery followed by chemoradiotherapy or radiotherapy is recommended by the National Comprehensive Cancer Network Guidelines. Metastatic and refractory thymic carcinomas require systemic pharmacotherapy. Combined carboplatin and paclitaxel, and cisplatin and anthracycline-based regimens have been shown a fair response rate and survival to provide a de facto standard of care when compared with other drugs employed as first-line chemotherapy. Cytotoxic agents have been pivotal for treating thymic carcinoma, as little is known regarding its tumorigenesis. In addition, genetic alterations, including driver mutations, which play an important role in treatments, have not yet been discovered. However, molecular pathways and biomarker studies assessing thymic epithelial tumors have been reported recently, resulting in the development of new agents, such as molecular targeted agents and immune checkpoint inhibitors. As treatment options are currently limited and the prognosis remains poor in metastases and recurrent thymic carcinoma, genetic alterations need to be assessed. In the present review, we focused on the current role of targeted therapies and immune checkpoint inhibitors in treating thymic carcinoma.

## 1. Introduction

Thymic carcinoma is a rare type of cancer derived from thymic epithelial cells. According to the RARECARE project supported by the European Commission, rare cancers have an annual incidence of less than 6 per 100,000 persons [1]. The incidence of thymic epithelial tumors is 0.15–0.32 cases per million; thymic carcinoma accounts for approximately 10–15% of the total [2]. According to the World Health Organization (WHO) classification, thymic tumors are classified into thymoma (major subtypes of type A, AB, B1, B2, and B3) and thymic carcinoma on the basis of clinical and immunological complications or tumor aggressiveness [3]. Thymic carcinoma can be classified into more than 10 subtypes, including squamous cell carcinoma, which comprises 70% of carcinomas and neuroendocrine tumors [4]. Type A and AB thymoma reportedly afford better prognoses than B1, B2, and B3 thymoma and carcinomas. In particular, thymic carcinoma has a poor prognosis, with a 5-year survival rate of 30–50% [5,6,7]. Prognosis depends on staging, which is commonly characterized by the Masaoka classification with Koga modification adopted by the International Thymic Malignancy Interest Group (ITMIG) [8]. Recently, ITIMG proposed a new staging system approved by the Union for International Cancer Control (UICC) in the eighth edition of the TNM classification [9]. Complete surgical resection is the standard treatment for thymic carcinoma; however, systemic chemotherapy is administered in metastatic or recurrent cases [10]. Given the rarity of the disease, therapeutic agents are often confirmed by results of small-sized phase II trials or retrospective studies. Platinum combination chemotherapy is widely employed as first-line chemotherapy for thymic carcinoma, with response rates ranging from 20% to 50% [11,12]. Cytotoxic chemotherapy remains a key treatment strategy for thymic carcinoma, possibly as biological behavior remains to be comprehensively elucidated and a lack of reported driver mutations. In addition, patients with thymoma and thymic carcinoma were formerly considered closely related. However, clinical presentations, more recent molecular markers, and genomic analysis point to more differences than similarities, and recent clinical studies have often evaluated these patients’ groups separately. Moreover, various targeted therapies and immune checkpoint inhibitors (ICIs) have been investigated and are discussed in the current review with the development of systemic treatment.

## 2. Cytotoxic Chemotherapy

Typically, cytotoxic chemotherapy for thymic carcinoma is based on chemotherapy administered for thymomas. Platinum-based combination chemotherapy plays an important role as first-line chemotherapy for advanced and recurrent thymic carcinomas. A carboplatin and paclitaxel regimen is an alternative for first-line therapy, as it resulted in the highest response rate in patients with thymic carcinomas in a phase II study, affording an overall response rate (ORR) of 36% and median progression-free survival (PFS) and overall survival (OS) of 7.5 months and not reached, respectively [13]. Another trial presented an ORR of 21.7%, along with a median PFS and OS of 5.0 and 20.0 months, respectively [14]. The association between platinum-based combination chemotherapy and anti-angiogenic agents has also been investigated. A phase II study evaluating the efficacy of combination ramucirumab, carboplatin, and paclitaxel remains ongoing (NCT03921671) (RELEVENT trial). Cisplatin and anthracycline-based regimens, such as CAP [15] (cisplatin, doxorubicin, and cyclophosphamide) or ADOC [16] (cisplatin, doxorubicin, vincristine, and cyclophosphamide), are regarded as key drugs. In addition, the ADOC regimen has demonstrated efficacy in treating thymic carcinoma (ORR of 50%); however, grade 3 or 4 leukopenia and neutropenia were observed in more than 70% of patients [13]. Furthermore, a phase II study assessing carboplatin and amrubicin showed a response rate of 30% for thymic carcinoma while presenting a rate of 42% in those who received no previous chemotherapy [17], thus indicating the potential of treatment options. However, in thymic carcinoma, no significant difference in response rates was observed between regimens containing anthracycline and non-anthracycline regimens using platinum-based chemotherapies [12]. The response rates after anthracycline-based and non-anthracycline-based chemotherapy for advanced thymic carcinoma were similar (41.8% vs. 40.9%; *p* < 0.91), whereas the response rates after cisplatin- and carboplatin-based chemotherapy for advanced thymic carcinoma differed significantly (53.6% vs. 32.8%; *p* = 0.0029) in 206 patients from 10 studies. Platinum with anthracycline-based chemotherapy is undoubtedly the optimal combination for advanced thymoma. For advanced thymic carcinoma, cisplatin-based chemotherapy may be superior to carboplatin-based chemotherapy.

The survival benefits of second-line cytotoxic chemotherapy for refractory thymic carcinoma remain unknown. According to the National Comprehensive Cancer Network (NCCN) guidelines, single-agent or non-platinum-based cytotoxic chemotherapy is recommended for recurrent thymoma and thymic carcinoma [18] Regarding cytotoxic chemotherapy, the guidelines fail to provide separate recommendations for thymic carcinoma and thymoma. For relapsed thymoma and thymic carcinoma, NCCN guidelines [18] recommend single-agent or non-platinum-based chemotherapy such as etoposide [19,20], gemcitabine [21], paclitaxel [22], ifosfamide [23], pemetrexed [24], 5-fluorouracil (5-FU) and leucovorin [25], and octreotide (including long-acting-formulation) plus prednisolone [26]. The NCCN guidelines do not provide separate recommendations for thymic carcinoma and thymoma [27]. According to retrospective studies, amrubicin [28], docetaxel [29], and S-1 [30,31,32] are reportedly effective for thymic carcinoma. In a phase II study of amrubicin [33] and pemetrexed [34] for recurrent thymic carcinoma, the ORRs were 11% and 9%, respectively. Recently, a phase II study of S-1 for recurrent thymic carcinoma showed clinical efficacy, presenting an ORR of 30.8% and median PFS and median OS of 4.3 months and 27.4 months, respectively [35]. In addition, an Italian research group has reported the efficacy of oral etoposide in patients with advanced platinum-pretreated thymic epithelial tumors [36]. Sixty patients had thymoma, and 40% had thymic carcinomas. Oral etoposide 50 mg daily for 3 weeks on and 1 week off every 28 days was administered. The authors recorded ORR and median PFS of 85% and 16 months (95%CI, 3–60), respectively, along with median PFS of 12 and 19 months for thymoma and thymic carcinoma, respectively. Etoposide inhibits angiogenesis both in vitro and in vivo by decreasing vascular endothelial growth factor (VEGF) production and microvessel density. The results suggest that angiogenesis inhibitors could effectively treat thymic carcinoma, considering lenvatinib [37] and sunitinib [38], which also inhibits angiogenesis, showed efficacy, which we discuss in Section 3.2. However, no objective response was observed in the study assessing bevacizumab, while multikinase inhibitors, such as lenvatinib and sunitinib, showed promising results.

## 3. Characterization of the Biology and Targeted Therapy

### 3.1. The Biology of Thymic Carcinoma

Compared with more common cancers, the biology of thymic epithelial tumors is poorly understood, given its rare occurrence. Biomarkers have been explored in thymic epithelial tumors, and targeted therapies have been developed for patients with pretreated thymic carcinoma. Immunohistochemical analysis revealed overexpression of certain genes. The main signaling pathways targeted for treating thymic malignancies are the epidermal growth factor receptor (EGFR), the KIT/mast/stem-cell growth factor receptor (KIT/SCFR), and insulin-like growth factor (IGF-1R) pathways. Anti-angiogenic agents have also been investigated. IGF-1R was detected by immunohistochemistry in 70% of cases [39]. On comparing thymoma and thymic carcinoma, EGFR overexpression was noted in 23% vs. 67–100%, HER2 overexpression in 6% vs. 53%, KIT overexpression in ˂5% vs. 73–86%, and BCL-2 overexpression in 14% vs. 100%, respectively [40]. Responses to available EGFR-tyrosine kinase inhibitors are yet to be reported; EGFR-activating mutations are found to infrequently occur in these tumors [41]. In contrast, EGFR and KIT mutations are uncommon [42]. The expression of KIT is reportedly associated with survival in thymic epithelial tumors [43], and KIT targeting regimens have been developed. As observed in other tumors, angiogenesis appears to play a pivotal role. Higher serum concentrations of VEGF and basic fibroblast growth factor (b-FGF) have been reported in patients with thymic carcinoma in comparison with those with thymoma [38]. The activation of the PI3K/AKT network plays a crucial role in thymic epithelial tumor growth and may sensitize thymic epithelial tumors to the inhibition of one of the key components of this intracellular axis [44]. We discuss these topics under Section 3.2.

The Cancer Genome Atlas project has revealed that the different histological subtypes of thymic epithelial tumors harbor specific molecular alterations, indicating that thymic cancer and thymoma exhibit distinct molecular profiles and major oncogenic pathways [45]. Whole exome sequencing was performed on 117 normal tumor pairs, and a high prevalence of GTF2I mutations and enrichment of mutations in HRAS, NRAS, and TP53 was identified. GTF2I is a thymoma-specific oncogene, and a high mutation frequency is seen in type A and AB thymomas. Clonality analyses revealed the potential of all four gene mutations occurring at onset or in the very early stages of tumor development. In thymic carcinoma, loss of chromosome 16q is common. The tumor mutation burden (TMB) of thymic epithelial tumors was the lowest average among adult cancers (an average of 0.48 mutations per megabase), and only two pediatric cancers presented a lower average TMB. However, the TMB of thymic carcinoma was significantly higher than that of thymoma.

In addition, thymic carcinoma carries a higher number of mutations, as well as recurrent mutations in known cancer-related genes, including TP53, CYLD, CDKN2A, BAP1, and PBRM1.

Recently, tuft cell-like cancer was described as a variant of small cell lung cancer that strongly depends on POU2F3 for survival in vitro [46], presenting a tuft cell-like gene expression signature but not tuft cell-like morphology. Tuft cells are epithelial chemosensory cells known to occur in the respiratory tract, intestine, and pancreas. A new molecular biological classification was proposed in 2019 that is based on four definitive molecules, namely, ASCL1, NEUROD1, YAP1, and POU2F3 [47]. These subtypes have been reported to exhibit different drug sensitivities in vitro. In the thymus, tuft cells help shape the microenvironment and influence innate immunity [48,49,50]. Molecular analysis using public databases and frozen tissues revealed that POU2F3 and GFI1B, both tuft cell-related genes, are highly expressed in thymic squamous cell carcinoma [51]. In addition, the authors assessed the KIT expression status; POU2F3 expression was strongly correlated with KIT expression, suggesting that POU2F3 may drive KIT expression.

### 3.2. The Role of Targeted Therapy

#### 3.2.1. KIT Inhibitor

KIT is a type III receptor tyrosine kinase expressed on the membrane of hematopoietic progenitor cells, mast cells, germ cells, and interstitial cells of Cajal (ICC) [52]. Stem cell factor (SCF), a ligand of KIT, induces receptor homodimerization, stimulating autophosphorylation, which downstream activates cell proliferation, differentiation, adhesion, and apoptosis [53]. KIT overexpression has been reported in thymic carcinoma, gastrointestinal stromal tumor (GIST), chronic myeloid leukemia (CML), mastocytosis, germ cell tumors such as seminoma, and malignant melanoma [43]. In thymic epithelial tumors, KIT expression is seen in 73–86% of thymic carcinomas, while its expression is ˂5% in thymoma [40]. Therefore, KIT is considered the classical diagnostic marker for thymic carcinoma, and the efficacy of targeted therapy for KIT has been reported.

The rate of KIT mutations remains low (9%) [40]. Mutations in c-KIT commonly occur in exons encoding functional domains of the tyrosine kinase receptor, namely, exons 9, 11, 13, and 17 [54], and these mutations are considered to be associated with survival [43]. In thymic carcinoma, clinical efficacy was reported for mutations in exon 11 V560del, exon 14 H697Y, exon 17 D820E, exon 11 V559G, exon 11 577–579del, exon 11 Y553N, and exon 13 K642E [41,55,56,57,58,59,60]. Exon 11 V560 del, V559G, and Y553N mutations were sensitive to imatinib [58,60,61]. D820E, H697Y, K642E, and del 557–559 were sensitive to sorafenib [41,55,57,59].

Imatinib inhibits receptors for Bcr-Abl, v-Abl, c-Abl, platelet-derived growth factor (PDGF), and c-KIT. Imatinib has been approved for CML, Philadelphia chromosome-positive acute lymphoblastic leukemia, myelodysplastic/myeloproliferative diseases, systemic mastocytosis, hypereosinophilic syndrome, dermatofibrosarcoma protuberans, and GIST. Imatinib was originally developed as a targeted therapy for CML as a Bcr-Abl tyrosine kinase inhibitor [62]. Imatinib also inhibits the tyrosine kinase activity of c-KIT, a membrane receptor tyrosine kinase. In most cases, GIST tumorigenesis has been attributed to the gain of function mutation of c-kit, and imatinib has reported clinical efficacy [63]. The efficacy of imatinib was investigated in two phase II trials for pretreated patients with thymic epithelial tumors; however, all patients failed to demonstrate a response [64,65]. One reason underlying the lack of response might be that c-KIT was evaluated in only 3 out of 15 patients, and no patient harbored a known c-KIT activating mutation [64]. In another phase II trial, no mutations were detected in the c-KIT or PDGFRA genes [65].

Sunitinib, a multi-target tyrosine kinase inhibitor including VEGF receptor (VEGFR), KIT, and PDGFR, showed a partial response (PR) in all four assessed patients with metastatic thymic cancers refractory to conventional therapies [66], indicating its efficacy. No mutations for c-KIT were detected, and one patient was refractory to imatinib in the previous treatment. A phase II study assessed sunitinib for pretreated thymic epithelial tumors, including 25 patients with thymic carcinoma and 16 with thymoma [38]. Sunitinib was administered as 50 mg orally once daily for 6-week cycles; a 6-week cycle consisted of 4 weeks of treatment followed by 2 weeks without treatment. The ORR, median PFS, and median OS were 26%, 7.2 months, and were not reached in patients with thymic carcinoma, respectively, while the ORR was only 6% in patients with thymoma. A study conducted in France using the RYTHMIC network prospective database reported an ORR of 22% (29% for thymoma and 20% for thymic cancer) and median PFS in the whole population of 3.7 months (5.4 months for thymoma and 3.3 months for thymic carcinoma) [67]. For improving tolerability, a phase II study assessed sunitinib at a modified dose of 50 mg once daily using a 2-weeks-on/1-week-off schedule [68]. However, only 8% of the patients with thymic carcinoma responded. 

Sorafenib is a multikinase inhibitor, including RAF, VEGFR, KIT, and PDGFR, and its efficacy has been reported for specific mutations [41,55,57,59]. For example, a case series of five patients with thymic carcinoma treated with sorafenib showed PR in two patients, stable disease (SD) in two patients, and progressive disease (PD) in one patient [69].

#### 3.2.2. Lenvatinib

Lenvatinib is a multi-targeted tyrosine kinase inhibitor of VEGFR, fibroblast growth factor receptors (FGFR), PDGFR-α, KIT, and RET, and has been approved for the treatment of unresectable thyroid cancer, hepatocellular carcinoma, and renal cell carcinoma. A single-arm, multicenter phase II study assessing lenvatinib (REMORA study) for unresectable advanced or metastatic thymic carcinoma that progressed following at least one platinum-based chemotherapy has been conducted [37]. Patients received 24 mg of lenvatinib orally once daily for 4-week cycles. Forty-two patients were enrolled, and the median follow-up period was 15·5 months (interquartile range (IQR) 13.1–17.5). The ORR and disease control rate (DCR) were 38% (90% CI 25.6–52.0) and 95% (90% CI 83.8–99.4), respectively, which met the primary endpoint. The median PFS and OS were 9.3 months (95% CI 7.7–13.9) and not reached (NR; 95% CI 16.1–NR), respectively, and the probability of 12-month PFS was 41% (95% CI 25.8–54.7), thus showing promising results. Additionally, post hoc subgroup analysis by histological type revealed ORRs of 46.7% (95% CI 25.8–65.7) and 16.7% (95% CI 2.1–48.4) for squamous cell carcinoma and non-squamous cell carcinoma, respectively. However, dose reduction was required in all cases, although the dose was determined on the basis of safety results from a previous study [70]. The most frequent grade 3 treatment-related adverse events were hypertension (27 (64%) of 42 patients) and palmar–plantar erythrodysesthesia syndrome (3 (7%)). No deaths due to adverse events were recorded. On the basis of the results of this trial, the country of Japan approved lenvatinib in March 2021. Some studies have suggested that the angiogenesis approach can enhance the antitumor activity of ICIs [71]. The combination of lenvatinib and pembrolizumab has been investigated in thyroid cancer, endometrial cancer, and renal cell cancer [72,73,74]. For thymic carcinoma, a phase II trial of this combination is currently ongoing (NCT03463460).

#### 3.2.3. mTOR Inhibitor 

The mammalian target of rapamycin (mTOR) is a serine-threonine kinase, a key component of the PI3K/AKT/mTOR intracellular axis. As mTOR regulates cell growth, proliferation, response to hypoxia, and tumor angiogenesis, it has been actively pursued as a therapeutic target in some cancers. The activation of PI3K/AKT reportedly plays an important role in cell proliferation in thymic epithelial carcinoma [75,76,77], and a phase II study of everolimus, an oral mTOR inhibitor, has been conducted [44]. The patients received oral everolimus 10 mg once daily. Fifty-one patients with advanced thymoma (*n* = 32) or thymic carcinoma (*n* = 19), who were eligible after the failure of at least one previous line of platinum-based chemotherapy, were enrolled. According to histology, PR was observed in three and two patients with thymoma and thymic carcinoma, respectively. The DCR was 88% (93.8% in thymoma and 77.8% in thymic carcinoma). The median PFS was 16.6 months for patients with thymoma and 5.6 months for those with thymic carcinoma. The reported adverse events did not differ from those described in other cancers [78,79]; however, three patients (6%) died of pneumonitis. Twenty-seven patients underwent immunohistochemical analysis and revealed significantly shorter survival in patients with tumor positivity for p4E-BP1 and IGF1R.

#### 3.2.4. Others

Tumor epithelial tumors are known to overexpress EGFR and VEGF, and studies have assessed EGFR and VEGF inhibitors. However, no remarkable efficacy has been reported [80,81]. Histone deacetylase (HDAC) regulates the expression of tumor suppressor genes, as well as the activities of transcriptional factors involved in cancer initiation and progression by altering either DNA or structural components of chromatin [82]. A phase II study has assessed the efficacy of belinostat, an HDAC inhibitor, in patients with recurrent or refractory thymic epithelial tumors [83]. Forty-one patients were enrolled, of whom 25 had thymoma and 16 had thymic carcinoma. Modest antitumor activity was observed, with PR in 2 patients, SD in 25 patients, and PD in 13 patients. No responses were documented among patients with thymic carcinoma.

Somatostatin (SST) receptors are reportedly expressed in various malignant tissues, including the thymus [84]. Octreotide, a somatostatin analog, has a high affinity for the SST receptor 2 [85] and is known to decrease the secretion of growth hormone and IGF-1 [86]. The expression of IGF-1R is often observed in thymic epithelial tumors, especially in patients with recurrent or advanced disease and aggressive histologic subtypes [39]. A phase II study has assessed octreotide alone or combined with prednisolone for advanced thymic epithelial tumors [26,87]. A study of 42 octreotide scan positive patients, of which 38 were fully assessable, revealed a complete response of 5.3% and PR of 25%; however, no responses were reported in patients with thymic carcinoma [26]. Clinical trials of targeted therapy are listed in Table 1.

### 3.3. Immunotherapy

#### 3.3.1. PD-L1 Expression in Thymic Cancers

Several studies have reported that thymic epithelial tumors exhibit high programmed death-ligand 1 (PD-L1) expression. PD-L1 is commonly expressed in 39.5–70% thymic carcinomas and 23–64% thymoma cases [88,89,90,91,92]. A study assessing 102 thymomas and 38 thymic carcinomas has reported that the expression of PD-L1 was not a significant negative factor for OS [89]. Another study reported that PD-L1 expression had no critical impact on OS [90]. Conversely, some studies suggest that PD-L1 expression may be a useful prognostic predictor. For example, a study assessing 25 patients with thymic carcinoma has reported that PD-L1 was highly expressed in 80% of patients, and low PD-L1 expression was a significant predictor of poor survival [91]. In contrast, another study involving surgical resection showed that PD-L1 expression was associated with worse PFS and OS [92]. Thus, it remains unclear as to whether PD-L1 levels could be useful for predicting prognosis. Differences in assays and evaluations among studies might be responsible for these results.

#### 3.3.2. The Efficacy of ICIs in Thymic Carcinoma

Several single-arm phase II studies have reported the presence of concerning anti-PD-1 antibodies. A study evaluating pembrolizumab in patients with recurrent thymic carcinoma has revealed an ORR of 22.5% (95% CI 10.8–38.5) in 40 eligible patients [93]. The median PFS, median OS, and 1-year PFS were 4.2 months, 24.9 months, and 29%, respectively. The most common grade 3 or 4 adverse events were increased aspartate aminotransferase and alanine aminotransferase (13% patients, each). Six patients (15%) developed severe immune-related adverse events, including two (5%) experienced myocarditis and polymyositis, requiring high-dose steroid therapy. The sample size of 40 was larger than that evaluated in other phase II trials for thymic carcinoma. Post hoc analyses of the association between response and 18-gene interferon-γ expression were performed in 33 patients; a significant correlation between the presence of the 18-gene interferon-γ signature and response was documented. In addition, PD-L1 expression was evaluated in 37 patients, and post hoc analysis demonstrated that the median PFS and OS were longer in patients with high PD-L1 expression. Moreover, a long-term follow-up of this study was reported [94]. With a median follow-up of 4.9 months, the median duration of response, median OS, and 5-year survival were 2.99 years, 2.12 years, and 18%, respectively. Another phase II trial assessed pembrolizumab in 33 patients with refractory or relapsed thymic epithelial tumors, of which 26 patients had thymic carcinoma. The ORR, median PFS, and median OS of thymic carcinoma were 19.2%, 6.1 months, and 14.5 months, respectively [95]. Five (71.4%) of seven patients with thymoma and four (15.4%) of 26 patients with thymic carcinoma reported grade 3 immune-related adverse events, including three patients (9.1%) with myocarditis. Adverse events in muscles and myocardium appear to be particularly frequent in thymic epithelial tumors when treating with ICI, which has been reported in other trials as well [93,96]. PD-L1 expression was assessed in 24 patients, and five patients with high PD-L1 expression showed PR, while none of the patients with low expression showed a response. For nivolumab, a phase II study evaluated 15 patients with recurrent or unresectable thymic carcinoma (PRIMER study). The ORR was 0%; however, DCR and median PFS were 73.3% and 3.8 months, respectively. This trial was stopped early for lack of responses as a predefined early termination criterion [97] (Table 2).

Compared with previous cytotoxic chemotherapies, the results of studies assessing pembrolizumab are notable, suggesting a possible survival benefit. Importantly, PD-L1 expression may predict the clinical benefit of pembrolizumab. A phase II study assessing sunitinib [38] has reported increased PD-L1 expression in circulating regulatory T cells and cytotoxic T-lymphocyte-associated protein 4 (CTLA-4) on circulating CD8+ T cells in most patients, which was associated with improved OS. However, the upregulation of immune checkpoint receptors, owing to T cell activation, may limit antitumor immunity in thymic epithelial tumors treated with sunitinib. Therefore, a combination of sunitinib and ICIs might enhance the response [38]. However, only one of eight patients previously treated with sunitinib responded to pembrolizumab, suggesting that PD-L1 expression might not always be an accurate biomarker.

Notably, thymic epithelial tumors reportedly possess the lowest TMB among all adult cancers [45]. Among these tumors, thymic cancer presents significantly higher TMB (*p* = 5.7 × 10^−5^) when compared with other thymoma histology. Typically, lower TMB can be associated with a lower response rate to ICIs, and thymic carcinoma showed an ORR of approximately 20%, as mentioned previously, which may be due to higher TMB than thymoma. CYLD, a tumor-suppressive gene, can be correlated with high PD-L1 expression in thymic cancer, with a mutation frequency of ˃10% [93]. The role of CYLD in PD-L1 expression was assessed by CYLD knockdown in tumor epithelial tumor cells; CYLD knockdown significantly enhanced PD-L1 expression in the presence of interferon (IFN)-γ stimulation in most cell lines, indicating that downregulation of CYLD promotes IFN-γ-mediated PD-L1 expression [98].

Currently, several clinical trials are assessing ICIs in patients with thymic carcinoma. To evaluate the efficacy of ICI monotherapy, phase II studies evaluating atezolizumab (NCT04321330), nivolumab (NCT03134118), and avelumab (NCT03076554) are ongoing. In addition, phase II studies evaluating combination therapy with a tyrosine kinase inhibitor and ICI are currently in progress, such as pembrolizumab plus sunitinib (NCT03463460) and pembrolizumab plus lenvatinib (PECATI) (NCT04710628). Table 3 presents ongoing clinical trials.

## 4. Discussion

Results of the comprehensive molecular analyses including the TCGA project have led to the recent developments in new systemic chemotherapies. However, cytotoxic chemotherapy is the best option for first-line therapy thus far. Since no significant difference in response rates was observed between regimens containing anthracycline and non-anthracycline regimens, a platinum-based antineoplastic drug seems to be the key drug. Carboplatin and paclitaxel regimen and carboplatin and amrubicin regimen have shown high response rates for thymic carcinoma. Cisplatin and anthracycline-based regimens have also reported high response rate. However severe adverse events were seen in more than 70% of patients received ADOC regimen, which indicates that some patients may not be tolerant to this regimen. For second- and upper-line therapy, targeted agents seem to be a prospective therapy. Lenvatinib has reported to have remarkable efficacy for thymic carcinoma. Although the sample size is small, sunitinib has also showed high response rate, which could become a promising alternative. Single cytotoxic agents have been used for patients who have progressed after first-line chemotherapy. In particular, S-1 and amrubicin are reported to have high response, which seems to be appropriate regimens.

ICIs against thymic carcinoma have yielded 0% to 22%. Second-line pembrolizumab has shown promising response. Responders also seem to have sustained benefit, which is reflected in an increase in the duration of response with long term follow-up. This suggests that pembrolizumab might be a good option for certain patients. However, future studies need to be designed for safety, and the discovery of biomarkers to identify patients at risk for of irAEs are necessary.

## 5. Conclusions

Recent discoveries of molecular markers concerning thymic carcinoma have aided in developing systemic therapies. Platinum-based chemotherapy is pivotal for systematic treatment, and promising results have been observed with targeted therapies and immunotherapies. In addition, several clinical trials assessing combination therapy and the development of new therapeutic agents are ongoing. However, no targetable mutations in a single gene have been detected to date. As thymic epithelial tumors are rare and histologically and molecularly heterogeneous, genomic profiling is often undertaken during the early stages of treatment. However, a further understanding of the underlying etiology is crucial and establishing a rare cancer network and customizing genomic panel for patients with thymic epithelial tumors might be necessary for future perspectives.

## Figures and Tables

**Table 1 cancers-13-05239-t001:** The efficacy of targeted therapy for thymic carcinoma in clinical trials.

Authors	Phase	Agents	Target	Number of Patients	ORR (%)	DCR (%)	PFS (Months)	OS (Months)
Palmieri et al. [64]	II	Imatinib	KIT	3	0	0	3 ^a^	NR
Giaccone et al. [65]	II	Imatinib	KIT	5	0	0	2 ^a^	4 ^a^
Thomas et al. [38]	II	Sunitinib	VEGFR, KIT, PDGFR, Flt-3	24	26	91	7.2	NR
Remon et al. [67]	retrospective	Sunitinib	VEGFR, KIT, PDGFR, Flt-3	20	20	55	3.3	12.3
Pagano et al. [69]	retrospective	Sorafenib	RAF, VEGFR, KIT, PDGFR,	5	40	80	6.4	21.2
Sato et al. [37]	II	Lenvatinib	VEGFR, FGFR, PDGFR, KIT, RET	42	38	95	9.3	NR
Zucali et al. [44]	II	Everolimus	mTOR	19	10.5	77.8	5.6	14.7
Kurup et al. [81]	II	Gefitinib	EGFR	7	0	NA	NA	NA
Bedano et al. [80]	II	Elrotinib, bevacizumab	EGFR, VEGF	7	0	NA	NA	NA
Giaccone et al. [83]	II	Belinostat	HDAC	16	0	50	2.7	12.4
Palmieri et al. [87]	II	Octreotide and PSL	Somatostatin receptor	6	37 ^a^	75 ^a^	14 ^a^	15 ^a^
Loehrer et al. [26]	II	Octreotide and/without PSL	Somatostatin receptor	6	0	67.1	4.5	23.4

^a^ Data of both thymic carcinoma and thymoma. Abbreviations: ORR, overall response rate; PFS, progression-free survival; OS, overall survival; NR, not reached; NA, not available.

**Table 2 cancers-13-05239-t002:** The efficacy of immunotherapy for thymic carcinoma in clinical trials.

Authors	Phase	ICI	Number of Patients	ORR (DCR) (%)	PFS (Months)	OS (Months)
Giaccone et al. [93,94]	II	Pembrolizumab	40	22.5 (75.0)	4.2	25.44
Cho et al. [95]	II	Pembrolizumab	24	19.2 (73.1)	6.1	14.5
Katsuya et al. [97]	II	Nivolumab	15	0 (73.3)	3.8	14.1

Abbreviations: ICI, immune checkpoint inhibitor; ORR, overall response rate; PFS, progression-free survival; OS, overall survival; DCR, disease control rate.

**Table 3 cancers-13-05239-t003:** Ongoing clinical trials for thymic carcinoma.

Study	Phase	Histology	Agents	Treatment Line	Primary Endpoint
NCT03921671	II	Thymic carcinoma, B3 thymoma	Ramucirumab, carboplatin, and paclitaxel	Untreated	ORR
NCT04321330	II	Thymic carcinoma	Atezolizumab	Pretreated with chemotherapy	ORR
NCT03134118	II	Thymic carcinoma, B3 thymoma	Nivolumab	Pretreated with platinum-based chemotherapy	PFS rate at month 6
NCT03076554	II	Thymic carcinoma, thymoma	Avelumab	Pretreated with platinum-based chemotherapy	Safety, ORR
NCT03463460	II	Thymic carcinoma	Pembrolizumab and sunitinib	Pretreated with platinum-based chemotherapy	ORR
NCT04710628	II	Thymic carcinoma, B3 thymoma	Pembrolizumab and lenvatinib	Pretreated with platinum-based chemotherapy	Median PFS

Abbreviations: PFS, progression-free survival; ORR, overall response rate.

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
