# Peer review of "Future Perspective of Chemotherapy and Pharmacotherapy in Thymic Carcinoma"

_cancers, 2021, doi:10.3390/cancers13205239_

Round 1

Reviewer 1 Report

This is a retrospective review of biology and therapy for thymic carcinoma which also outlines ongoing studies.

Overall comments:

  • This is a comprehensive review. As this is primarily focused on thymic carcinoma, the authors can minimize descriptions on thymoma, except where there are important comparisons. (this might be benefitted with a table). For instance, a paragraph (lines 165-68) on GTF2I in thymoma has no relevance in this paper on thymic carcinoma.
  • Another area (line 360), where the authors should dissect and clarify a bit more about TMB in Thymic carcinoma from the TCGA reference. The TMB was significantly higher (P=5.7x 10-5) for TC compared to thymoma. By conflating all the TETs together it can mislead the casual reader.
  • As another example to this, lines 362-64, proport surprise that “Typically, lower TMB can be associated with a lower response rate to ICIs…”, but if taken with the above noted, this may explain why it has activity
  • Finally, for the conclusion, the paper doesn’t unfold to me a better “understanding of the biology” but rather points to the discovery of some molecular markers that have steered development of therapies.

Minor points:

  • Line 28 needs editing “? and remain”
  • Line 61. Very awkward sentence. Might consider that “T and TC were formerly considered closely related, clinical presentations, more recent molecular markers and genomic analysis point to more differences than similarities.” Or something to that effect.
  • Lines 139-41. I believe that the authors have transposed the numbers for T and TC respectively.
  • Line 160. The TCGA paper found high proportion of 16q loss, not “16q7” loss.
  • Line 195. Would move the sentence to first line of the next paragraph

Table 1. I found confusing with ORR (DCR) (%) would either just do ORR with % or can have another column with DCR%. 

Author Response

RESPONSES TO THE REVIEWERS

Responses to Reviewer #1

Overall comments
Comment 1: This is a comprehensive review. As this is primarily focused on thymic carcinoma, the authors can minimize descriptions on thymoma, except where there are important comparisons. (this might be benefitted with a table). For instance, a paragraph (lines 165-68) on GTF2I in thymoma has no relevance in this paper on thymic carcinoma.

Response: Thank you for your valuable comments. As you have mentioned, minimizing the thymoma section is needed. We have deleted the paragraph on GTF2I in thymoma (page 4, line 182). We have also deleted the sentences concerning cytotoxic chemotherapy on thymoma (page 3, line 100).

Comment 2: Another area (line 360), where the authors should dissect and clarify a bit more about TMB in Thymic carcinoma from the TCGA reference. The TMB was significantly higher (P=5.7x 10-5) for TC compared to thymoma. By conflating all the TETs together it can mislead the casual reader.

Response: Thank you for your valuable comments. We have changed the text and added the information that The TMB was significantly higher (P=5.7x 10-5) for thymic carcinoma compared to thymoma (page 9, lines 424–425).

Comment 3: As another example to this, lines 362-64, proport surprise that “Typically, lower TMB can be associated with a lower response rate to ICIs…”, but if taken with the above noted, this may explain why it has activity.

Response: Thank you for your comments. We have changed the sentence to ‘Typically, lower TMB can be associated with a lower response rate to ICIs, and thymic carcinoma showed an ORR of approximately 20%, as mentioned previously, which may be due to higher TMB than thymoma.’ (page 9, line425-428)

Comment 4: Finally, for the conclusion, the paper doesn’t unfold to me a better “understanding of the biology” but rather points to the discovery of some molecular markers that have steered development of therapies.

Response: Thank you for your comments. We have changed the first sentence of the conclusion to ‘Recent discoveries of molecular markers concerning thymic carcinoma have aided in developing systemic therapies’ (page 10, lines 473–474).

Minor

Comment 1: Line 28 needs editing “? and remain”

Response: Thank you for pointing out our mistakes. We have deleted ‘remain exclusive’ (page1, line 29).

Comment 2: Line 61. Very awkward sentence. Might consider that “T and TC were formerly considered closely related, clinical presentations, more recent molecular markers and genomic analysis point to more differences than similarities.” Or something to that effect.

Response: Thank you for your comments. We have changed the sentence to ‘patients with thymoma and thymic carcinoma were formerly considered closely related. However, clinical presentations, more recent molecular markers and genomic analysis point to more differences than similarities, and recent clinical studies have often evaluated these patients’ groups separately’ (page 2, lines 66-69).

Comment 3: Lines 139-41. I believe that the authors have transposed the numbers for T and TC respectively.

Response: Thank you for your comment. We have reconfirmed the numbers of EGFR, HER2, KIT and BCL-2 overexpression of T and TC reported in reference 40, and the numbers were correct (page 3, line146-148).

Comment 4: Line 160. The TCGA paper found high proportion of 16q loss, not “16q7” loss.

Response: Thank you for pointing out our mistakes. We have corrected to ’loss of chromosome 16q’ (page4, line 178).

Comment 5: Line 195. Would move the sentence to first line of the next paragraph.

Response: We have move the sentence in line 95 to the next paragraph as you suggested.

Comment 6: Table 1. I found confusing with ORR (DCR) (%) would either just do ORR with % or can have another column with DCR%.

Response: Thank you for your comments. We have made another column for DCR (Table 1).

Reviewer 2 Report

In this review, the authors give a broad overview of therapeutic options in relapsed and / or metastatic thymic carcinoma requiring systemic therapy

Major point:

The manuscript thoroughly and accurately reviews the evidence for the respective treatments in thymic carcinoma. The review closely adheres to the published data in detail. On the other hand, no treatment algorithms are proposed or discussed. So the authors do not address the weighing of the different options for second line treatment (altered chemotherapy regimen versus Levantinib versus Pembrolizumab). Obviously, there is no head to head data or evidence for weighing these options, and the lenvatinib and pembrolizumab trials only specify the number of previous lines, but not responses depending on a previous second line treatment. Nevertheless, the authors should make an attempt to discuss ideal therapy sequence from the perspective of their experience. This could open a new angle beyond the mere reviewing of response data, and would make the review more attractive for the broad readership.   

The systematic review “Systemic treatments for thymoma and thymic carcinoma: A systematic review by Berghmans et al (Lung Cancer 2018) should be considered, since the focus of both reviews is similar and the systematic approach by Berghmans et al. is helpful with weighing the different chemotherapy options

Minor points:

Abstract: Line 24: Carboplatin / Paclitaxel not as generally regarded as standard of care as implied, since also platin / anthracycline combinations are considered as first line treatment (with higher ORR rates); this equally applies to lines 69-73 of the review  

Abstract: Line 27-29: The sentence “In addition … remain elusive” appears an error, does not make sense grammatically or logically.  

Line: 205-215: the description of efficacy of imatinib in non-thymic carcinoma entities appears overly long and - being out of focus - should be shortened

Regarding point 3.3.2 (efficacy of ICIs in thymic carcinoma) the increased risk of side effects with immune checkpoint inhibition in the thymic carcinoma entity should be mentioned as well (e.g. ref 97).

Line 343-4: here, it should be made clearer that the trial (ref 100) was terminated early due to lack of responses, e.g. “was stopped early for lack of responses as a predefined early termination criterion” 

Author Response

Responses to Reviewer #2

Major
Comment 1: The manuscript thoroughly and accurately reviews the evidence for the respective treatments in thymic carcinoma. The review closely adheres to the published data in detail. On the other hand, no treatment algorithms are proposed or discussed. So the authors do not address the weighing of the different options for second line treatment (altered chemotherapy regimen versus Levantinib versus Pembrolizumab). Obviously, there is no head to head data or evidence for weighing these options, and the lenvatinib and pembrolizumab trials only specify the number of previous lines, but not responses depending on a previous second line treatment. Nevertheless, the authors should make an attempt to discuss ideal therapy sequence from the perspective of their experience. This could open a new angle beyond the mere reviewing of response data, and would make the review more attractive for the broad readership. The systematic review “Systemic treatments for thymoma and thymic carcinoma: A systematic review by Berghmans et al (Lung Cancer 2018) should be considered, since the focus of both reviews is similar and the systematic approach by Berghmans et al. is helpful with weighing the different chemotherapy options

Response: Thank you for your valuable comments. As you mentioned, the attempt to discuss the ideal therapy is important for this review. We have added ‘4. Discussion’ section for this.

Minor

Comment 1: Abstract: Line 24: Carboplatin / Paclitaxel not as generally regarded as standard of care as implied, since also platin / anthracycline combinations are considered as first line treatment (with higher ORR rates); this equally applies to lines 69-73 of the review.

Response: Thank you for your valuable comment. As you have pointed out, platin / anthracycline combinations are considered as first line treatment as well. We have changed the abstract to ‘Combined carboplatin and paclitaxel, and cisplatin and anthracycline-based regimens have been shown a fair response rate and survival to provide a de facto standard of care when compared with other drugs employed as first-line chemotherapy (page 1, line 24-26).’ We have also changed the review to ‘A carboplatin and paclitaxel regimen is an alternative for first-line therapy,’ (page 2, line 77).

Comment 2: Abstract: Line 27-29: The sentence “In addition … remain elusive” appears an error, does not make sense grammatically or logically.

Response: Thank you for pointing out our mistakes. We have deleted ‘remain exclusive’ (page1, line 29).

Comment 3: Line: 205-215: the description of efficacy of imatinib in non-thymic carcinoma entities appears overly long and - being out of focus - should be shortened

Response: Thank you for your valuable comment. We have shortened the text by deleting, ‘In leukemia cells, the Philadelphia chromosome encodes the Bcr-Abl tyrosine kinase protein expressed in the cytoplasm, and research assessing specific inhibitors of this tyrosine kinase as a potential molecular target for cancer led to the development of imatinib’ (page5, line 234).

Comment 4: Regarding point 3.3.2 (efficacy of ICIs in thymic carcinoma) the increased risk of side effects with immune checkpoint inhibition in the thymic carcinoma entity should be mentioned as well (e.g. ref 97).

Response: Thank you for your suggestions. We have added the information of the side effects of ICIs in section 3.3.2. (page 8, lines 376–379) (page 8, lines 391–395).

Comment 5: Line 343-4: here, it should be made clearer that the trial (ref 100) was terminated early due to lack of responses, e.g. “was stopped early for lack of responses as a predefined early termination criterion”

Response: Thank you for your valuable comment. We have changed the sentence to ‘This trial was stopped early for lack of responses as a predefined early termination criterion’(page 8, line 399-400).